# DiffS2UT: A Semantic Preserving Diffusion Model for Textless Direct Speech-to-Speech Translation

**Yongxin Zhu**[1,3], **Zhujin Gao**[2,3], **Xinyuan Zhou**[4], **Zhongyi Ye**[4], **Linli Xu**[2,3]

[1]School of Data Science, University of Science and Technology of China
[2]School of Computer Science and Technology, University of Science and Technology of China
[3]State Key Laboratory of Cognitive Intelligence
[4]iFlytek Research
{zyx2016, gaozhujin}@mail.ustc.edu.cn, linlixu@ustc.edu.cn,
{xyzhou15, zyye7}@iflytek.com

## Abstract

While Diffusion Generative Models have achieved great success on image generation tasks, how to efficiently and effectively incorporate them into speech generation especially translation tasks remains a non-trivial problem. Specifically, due to the low information density of speech data, the transformed discrete speech unit sequence is much longer than the corresponding text transcription, posing significant challenges to existing auto-regressive models. Furthermore, it is not optimal to brutally apply discrete diffusion on the speech unit sequence while disregarding the continuous space structure, which will degrade the generation performance significantly. In this paper, we propose a novel diffusion model by applying the diffusion forward process in the *continuous* speech representation space, while employing the diffusion backward process in the *discrete* speech unit space. In this way, we preserve the semantic structure of the continuous speech representation space in the diffusion process and integrate the continuous and discrete diffusion models. We conduct extensive experiments on the textless direct speech-to-speech translation task, where the proposed method achieves comparable results to the computationally intensive auto-regressive baselines (500 steps on average) with significantly fewer decoding steps (50 steps).

## 1 Introduction

Speech-to-speech translation (S2ST) aims at translating speech of one language to speech of another language, breaking the communication barriers between people around the world who speak different languages. Conventional cascaded systems (Lavie et al., 1997; Nakamura et al., 2006) for S2ST typically consist of automatic speech recognition (ASR), machine translation (MT) or end-to-end speech-to-text translation (S2T), followed by text-to-speech synthesis (TTS). However, integrating multiple separated modules into a single system would cause the problem of error propagation and incur expensive computational costs. On one hand, text-based cascaded systems face challenges when dealing with low-resource languages that lack text annotations or even without written systems (Chen et al., 2022a). On the other hand, the speech-to-text process disregards acoustic information such as accentuation, attitude, and emotional nuances, which are crucial for effective human spoken communication.

Recent works of textless direct S2ST models (Tjandra et al., 2019; Kano et al., 2021; Jia et al., 2019, 2022; Zhang et al., 2021; Lee et al., 2021, 2022; Popuri et al., 2022; Wei et al., 2022) directly translate speech to speech in an end to end manner without the intermediate generation steps relying on text transcripts. Among them, S2UT model (Lee et al., 2021, 2022) takes the advantage of recent progress in spoken language modeling (Lakhotia et al., 2021) to obtain the discrete speech representations, or discrete units, to build the textless S2ST systems. The entire system consists of a sequence-to-sequence transformer (Vaswani et al., 2017) with a speech encoder and an auto-regressive unit decoder, followed by a unit HiFi-GAN vocoder (Polyak et al., 2021) trained separately for unit-to-waveform conversion. Experiments (Chen et al., 2022a) have demonstrated notable improvements in translation for unwritten languages. Despite the promising translation capabilities of the aforementioned auto-regressive-based S2ST models, they suffer from $O(N)$ decoding time complexity for predicting one token per step. Furthermore, the learned discrete speech unit sequences are typically much longer than the corresponding text transcripts due to the low information density of speech data, with a rate of 50 units per second (sequence length up to 500 on average), where the generation latency bottleneck is exposed.

In the meantime, diffusion generative models (Sohl-Dickstein et al., 2015; Ho et al., 2020; Song

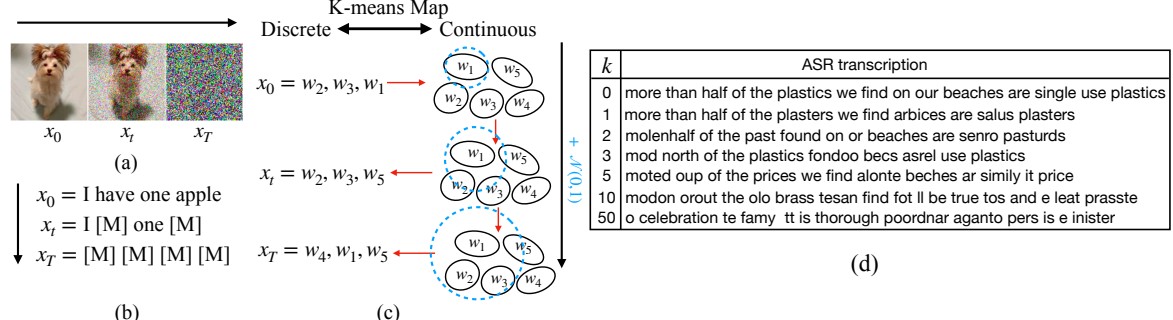

Figure 1: Illustration of different diffusion forward processes. (a) Continuous diffusion on images. The image is corrupted gradually by the Gaussian noise. (b) Discrete absorbing diffusion on text sequences. The text sequence is corrupted with masks. (c) Our proposed diffusion process. The discrete units are converted between the continuous space and the discrete space by K-means mapping. The Gaussian noise is added in the continuous space. (d) The ASR transcription of the noisy speech converted from the perturbed units. The units are perturbed by replacing them with the $k$-NN search in the continuous K-means space.

et al., 2020) have made remarkable advancements in generating high-quality images (Rombach et al., 2022), audios (Popov et al., 2021; Shen et al., 2023), and texts (Austin et al., 2021; Hoogeboom et al., 2021). Diffusion generative models are typically defined as a forward process and a backward process. The forward process gradually corrupts the data with noise, while the backward process starts with the pure noise and performs ancestral sampling to denoise the corrupted data. We compare different diffusion forward processes for specific data structures in Figure 1. For continuous data like audios or images in Figure 1(a), current diffusion models (Ho et al., 2020) typically add Gaussian noise gradually to the pixel space or audio spectrogram space (Popov et al., 2021). For discrete data like texts in Figure 1(b), current diffusion models (Austin et al., 2021; Hoogeboom et al., 2021) perturb the data with absorbing or multinomial transition matrices.

However, speech is a unique modality that lies between audio and text. On one hand, speech is a continuous form of acoustic data represented as a waveform. On the other hand, speech encompasses high-level linguistic information similar to discrete text data and can be represented as discrete units with SSL models, which is crucial for the translation task. It is challenging to directly apply the diffusion forward process like images or texts for S2ST. For example, Translatoron (Jia et al., 2019) directly generates the continuous spectrogram for speech translation but the performance is significantly inferior to the discrete unit based model S2UT (Lee et al., 2021). We conduct experiments similar to DiffSound (Yang et al., 2023)

that uses naive discrete diffusion models to generate discrete units for S2ST. Our primary results (see Table 1) demonstrate that it is sub-optimal to brutally apply discrete diffusion on the speech unit sequences, which would degrade the generation performance by a large margin. Typically, the vocabulary for the text translation task is constructed by BPE (Sennrich et al., 2016), and the semantics of the tokens require a large amount of training data to learn. In contrast, discrete speech units are acquired by the K-means clustering algorithm in the continuous speech representation space learned by the self-supervised learning models. Consequently, structural relationships exist among these units naturally. As shown in Figure 1(d), we perturb discrete speech units by replacing them with $k$-NN search in the continuous K-means space, and then convert them to the waveform with the vocoder described in Section 3.2. We transcribe the noisy speech with the open-sourced ASR system to "see" the pronunciation. We find that when $k$ is small, the noisy speech *sounds* similar to the original speech. The results show that the close K-means centroids share similar acoustical meanings.

To tackle these challenges, in this work, we propose a new paradigm called DiffS2UT for the S2ST task, a novel diffusion generative model that bridges the continuous representation space and discrete unit space of speech, while simultaneously preserving the semantic structure of the continuous speech representation space as well. Specifically, as depicted in Figure 1(c), there exists an one-to-one map between the discrete units and the continuous K-means centroid vectors. Therefore, in the diffusion forward process, we first map a unit

to the continuous vector derived from the learned K-means space. Next, we gradually add Gaussian noise to corrupt it with the continuous diffusion process. Subsequently, the noisy embedding vector is converted into the discrete unit index through the nearest neighbor search. As a consequence, we obtain the perturbed discrete unit sequences which are derived from the semantically structured continuous space, rather than being uniformly sampled from the vocabulary as in (Hoogeboom et al., 2021). The perturbed unit sequences are semantically close to the unperturbed ones when the added noise is small, which cannot be achieved in the discrete diffusion process. In the diffusion backward process, we train the model to predict the uncorrupted discrete unit sequences with cross-entropy in parallel.

We evaluate our framework DiffS2UT on the real-world S2ST datasets (Wang et al., 2021; Iranzo-Sánchez et al., 2020). The model achieves 14.8/15.2/14.5/13.6 BLEU score on the Eurapal-ST Es-En/Fr-En/En-Es/En-Fr test sets, significantly surpassing the vanilla diffusion models, and achieving comparable performance to auto-regressive models while requiring much fewer generation steps.

To summarize, the main contributions of this paper are:

- To the best of our knowledge, this is the first work that effectively introduces diffusion generative models to the textless S2ST task. The model significantly accelerate the generation process (50 steps v.s. 500 steps on average), while maintaining the generation quality comparable to the auto-regressive models.

- We propose a novel diffusion framework that integrates the continuous and discrete diffusion models by decoupling the diffusion forward and backward processes.

- The proposed diffusion model DiffS2UT bridges the continuous and discrete spaces of speech and preserves the semantic structure of the speech representation. Experimental results demonstrate its superiority over vanilla diffusion models.

## 2 Related Work

**Textless Direct Speech to Speech Translation**
Textless direct speech-to-speech translation (S2ST) aims at translating the source speech to the target speech directly without any text intermediates. Translatoron (Jia et al., 2019, 2022) is the first S2ST model to translate speeches by generating spectrograms directly. Tjandra et al. (2019); Zhang et al. (2021) build the S2ST system by pre-training a VQ-VAE model to convert target speeches into discrete codes and learn a speech-to-code translation model. S2UT model (Lee et al., 2021) utilizes the self-supervised speech encoder HuBERT (Hsu et al., 2021) pre-trained on large corpus of unlabeled speeches to convert speeches into discrete units with K-means clustering, which outperforms the VQ-VAE based approach in (Zhang et al., 2021). Lee et al. (2022) extend the S2UT model to the real-word S2ST data VoxPopuli-S2S (Wang et al., 2021) and propose a speech normalizer to obtain the speaker-invariant representations. Duquenne et al. (2022) propose a large-scale multilingual S2ST corpus with the speech mining method (Duquenne et al., 2021). By introducing the pre-trained models including the wav2vec (Baevski et al., 2020) encoder and mBART (Liu et al., 2020) decoder to the S2UT model, the translation quality is further boosted in (Popuri et al., 2022). TranSpeech (Huang et al., 2023) alleviates the acoustic multi-modal problem with bilateral perturbation. Speech2S (Wei et al., 2022) performs joint pre-training on speech and text datasets to align the acoustic and textual modalities. Although the above methods achieve good performance in the S2ST task, they are built on the auto-regressive generation models and suffer from the slow generation speed due to the excessive lengths of speech sequences. Our work leverages diffusion generative models to speed up the translation process without compromising the translation quality.

**Diffusion Probabilistic Models** Diffusion generative models (Sohl-Dickstein et al., 2015; Ho et al., 2020; Song et al., 2020) have achieved remarkable progress in generating high-quality images (Rombach et al., 2022), audios (Popov et al., 2021; Shen et al., 2023), and texts (Austin et al., 2021; Hoogeboom et al., 2021). DDPM (Ho et al., 2020) defines the generative process as the reverse of a particular Markovian diffusion process. By gradually corrupting an image with Gaussian noise and training a model based on U-Net to denoise it, DDPM can generate high quality images. Multinomial diffusion (Hoogeboom et al., 2021) proposes a uniform corruption process for categorical variables,

which is extended by D3PMs (Austin et al., 2021) to support general transition matrices, including an absorbing variant that draws close connections to masked LMs. Recent works (Gong et al., 2022; Gao et al., 2022) aim at conducting the Gaussian diffusion process over token embeddings so that the continuous diffusion models can be applied to discrete texts. Some works (Chen et al., 2022b) consider converting discrete tokens to bit strings and model them as real values. However, these approaches neglect the semantic relationships between tokens because the vocabulary is constructed using BPE, limiting the model performance. In this work, we introduce the diffusion generative models into the S2ST task. We conduct a thorough investigation of the unique properties of the speech modality and propose a novel diffusion paradigm decoupling the diffusion forward and backward process to bridge the discrete and continuous spaces of speech, while preserving the semantic structure of speech representations.

## 3 Background

In this section, we start with the formulations of both continuous and discrete diffusion generative models. Then we introduce some preliminaries for the proposed DiffS2UT system, such as how to convert a speech to discrete units and units to a waveform.

### 3.1 Diffusion Generative Models

The diffusion generative models are typically defined as a forward process and a backward process. Given the continuous data $\mathbf{x}_0 \sim p_{data}(\mathbf{x}_0) \in \mathbb{R}^d$, the continuous diffusion forward process usually perturbs it into the Gaussian noise $\mathbf{x}_T \sim \mathcal{N}(\mathbf{0}, \mathbf{I})$ through a series of latent variables $\mathbf{x}_1, \ldots, \mathbf{x}_T$ in a Markov transition:

$$q(\mathbf{x}_t|\mathbf{x}_{t-1}) = \mathcal{N}(\mathbf{x}_t; \sqrt{1 - \beta_t}\mathbf{x}_{t-1}, \beta_t\mathbf{I}), \quad (1)$$

where $\beta_t$ controls the noise level added at the timestep $t$. For any arbitrary $t$, sampling $\mathbf{x}_t$ from $\mathbf{x}_0$ can be achieved in a closed form with the reparameterization trick:

$$q(\mathbf{x}_t|\mathbf{x}_0) = \mathcal{N}(\mathbf{x}_t; \sqrt{1 - \bar{\beta}_t}\mathbf{x}_{t-1}, \bar{\beta}_t\mathbf{I}), \quad (2)$$

where $\bar{\beta}_t := 1 - \prod_{i=0}^{t}(1 - \beta_i)$. In the backward process, the model learns to recover $\mathbf{x}_0$ by denoising from $\mathbf{x}_t$ step by step:

$$p_\theta(\mathbf{x}_{t-1}|\mathbf{x}_t) = \mathcal{N}(\mathbf{x}_{t-1}; \boldsymbol{\mu}_\theta(\mathbf{x}_t, t), \boldsymbol{\Sigma}_\theta(\mathbf{x}_t, t)), \quad (3)$$

where $\boldsymbol{\mu}_\theta, \boldsymbol{\Sigma}_\theta$ are the predicted mean and variance of $q(\mathbf{x}_{t-1}|\mathbf{x}_t)$, $\theta$ denotes the model parameters. Following the derivation in Ho et al. (2020), $\boldsymbol{\Sigma}_\theta$ is set as $\sigma_t^2\mathbf{I}$ and the objective function of the diffusion model can be written as a simplified variational lower-bound (VLB) of $\log p_\theta(\mathbf{x}_0)$:

$$\mathcal{L}_{\text{vlb}} = \mathbb{E}_{\mathbf{x}_0,t}[\|\boldsymbol{\mu}_\theta(\mathbf{x}_t, t) - \hat{\boldsymbol{\mu}}(\mathbf{x}_t, \mathbf{x}_0)\|^2] \quad (4)$$

For the discrete data represented as one-hot vectors $\mathbf{x}_0 \sim p_{data}(\mathbf{x}_0) \in \{0, 1\}^K$, the discrete diffusion forward process usually involves modeling noise through multinomial transition (Hoogeboom et al., 2021) and absorbing transition (Austin et al., 2021). The multinomial transition adopts a uniform noise distribution over the vocabulary $\{1, \ldots, K\}$; alternatively, the absorbing transition specifies noise to be a point mass with all of the probability on an absorbing state. In the backward process, the model $p_\theta$ is adopted to predict $\mathbf{x}_0$ with a softmax function, representing the probability distribution for each token. The objective function of the discrete diffusion model can be written as the negative log-likelihood loss:

$$\mathcal{L}_{\text{vlb}} = \mathbb{E}_{\mathbf{x}_0,t} - [\mathbf{x}_0 \log p_\theta(\mathbf{x}_t, t)], \quad (5)$$

### 3.2 Pre-processing and Post-processing for DiffS2UT

We describe the speech-to-unit method and the unit-to-speech method below.

**Speech to Unit** Lee et al. (2022) pre-train Multilingual HuBERT (mHuBERT) (Hsu et al., 2021) on the data combined of En, Es and Fr from the unlabeled speech dataset VoxPopuli (Wang et al., 2021), which contains 4.5k hours of data for En, Es and Fr, respectively. We follow Lee et al. (2022) and use mHuBERT to discretize the target speech. As shown in the left of Figure 2(a), the speech-to-unit discretization method uses the last iteration of the mHuBERT model for feature extraction, followed by the K-means clustering algorithm. The learned K cluster centroids are used to convert the audio into a sequence of cluster indices, which are referred to as units.

**Unit to Speech** As illustrated in the right of Figure 2(a), the unit-to-speech conversion is done with the discrete unit-based HiFi-GAN vocoder proposed in (Polyak et al., 2021). The vocoder is trained separately from the S2UT model, with the combination of the generator-discriminator loss

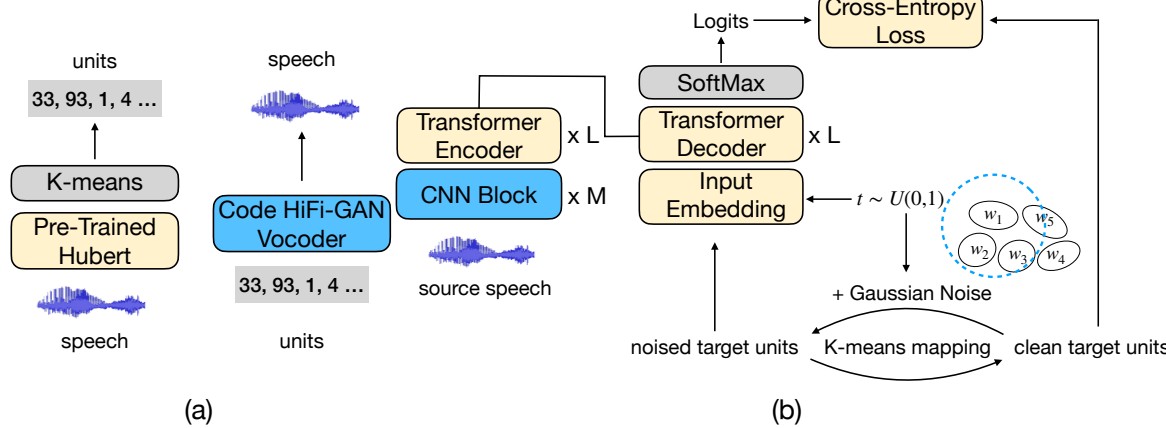

Figure 2: Illustration of the model architecture. (a) Left: The speech-to-unit model with mHuBERT and K-means to discretize the speech input. Right: The unit-based HiFi-GAN vocoder for unit-to-speech conversion. (b) Our proposed DiffS2UT translation model. The speech units are transformed and perturbed between the discrete space and continuous space for diffusion.

from HiFi-GAN (Kong et al., 2020) and MSE of the predicted duration of each unit in the logarithmic domain.

## 4 DiffS2UT

In this section, we elaborate our proposed model DiffS2UT in detail. We first describe the model architecture for S2ST, followed by the novel training and sampling algorithms for diffusion.

### 4.1 Model Architecture

Figure 2(b) depicts the overall architecture of the DiffS2UT system. The entire system consists of a speech encoder and a unit decoder. The speech encoder is constructed using a CNN-based speech down-sampling module and a stack of Transformer encoder blocks (Vaswani et al., 2017). The unit decoder consists of an input embedding layer that embeds discrete units and diffusion time steps, followed by a stack of Transformer decoder blocks. Given the source speech $\mathbf{c}$, the model learns to generate the target unit sequence $\mathbf{x} = [x_1, \ldots, x_n]$ by estimating $p_\theta(\mathbf{x}|\mathbf{c})$. Previous models factorize the unit sequence in an auto-regressive manner and estimate the units from left to right, i.e., $p_\theta(\mathbf{x}|\mathbf{c}) = \prod_{i=1}^n p_\theta(x_i|x_{<i}, \mathbf{c})$. In contrast, the diffusion decoder is trained with the non-causal attention mask and estimates all tokens simultaneously, which means each token can not only attend to the tokens before it but the tokens after, i.e., $p_\theta(\mathbf{x}|\mathbf{c}, t) = \prod_{i=1}^n p_\theta(x_i|\mathbf{c}, t)$. We denote the K-means mapping function as $g$, which can convert the discrete unit indices to the continuous K-means

cluster centroid vectors. And the inversion function $g^{-1}$ works by conducting the nearest neighbor search in the K-means Euclidean space, converting the continuous vector to the K-means cluster indices (units).

For each unit sequence as the decoder input, we first convert it to the continuous space that encapsulates the semantic structure learned by SSL models and K-means. Then we perturb it with the Gaussian noise as in the continuous diffusion forward process, followed by the inversion function converting it back to discrete units. At last, the model learns to recover the correct unit sequence with the corrupted units and time step $t$ as the input. The model is optimized with negative log-likelihood objective.

### 4.2 Training

The training details are described in Algorithm 1. Following standard practices in diffusion models, we randomly sample a time step $t$ and optimize the model parameter $\theta$ with respect to the objective $\mathcal{L}(\theta)$. In our case, $\mathcal{L}(\theta) = -\mathbf{x}_0 \log p_\theta(\mathbf{x}_t, t)$, which is the KL divergence of $\mathbf{x}_0$ and $p_\theta(\mathbf{x}_t, t)$. Given a unit sequence $\mathbf{x}_0 \sim p_{data}(\mathbf{x}_0) \in \{1, \ldots, K\}^n$, we first convert it to the continuous speech representation $\mathbf{v}_0 = g(\mathbf{x}_0) \in \mathbb{R}^{n \times d}$ with the K-means mapping function $g$. Then we sample $\mathbf{v}_t$ by perturbing $\mathbf{v}_0$ with the Gaussian noise as in continuous diffusion models

$$q(\mathbf{v}_t|\mathbf{v}_0) = \mathcal{N}(\mathbf{v}_t; \sqrt{1 - \bar{\beta}_t}\mathbf{v}_{t-1}, \bar{\beta}_t\mathbf{I}). \quad (6)$$

After that we transform $\mathbf{v}_t$ back to the discrete

**Algorithm 1** Training DiffS2UT

**Input:** network $f_\theta$, data distribution $p_{data}(\mathbf{x}_0)$, and K-means mapping function $g$.
**Output:** model parameters $\theta$.
**repeat**
    Draw $\mathbf{x}_0 \sim p_{data}(\mathbf{x}_0)$;
    Draw $t \in \text{Uniform}(\{1, \ldots, T\})$;
    Convert discrete $\mathbf{x}_0$ to continuous space $\mathbf{v}_0 = g(\mathbf{x}_0)$;
    Draw $\mathbf{v}_t \sim q(\mathbf{v}_t|\mathbf{v}_0)$;
    Convert continuous $\mathbf{v}_t$ to discrete space $\mathbf{x}_t = g^{-1}(\mathbf{v}_t)$;
    $\mathcal{L}(\theta) = -\mathbf{x}_0 \log p_\theta(\mathbf{x}_t, t)$;
    Minimize $\mathcal{L}(\theta)$ with respect to $\theta$;
**until** converged

---

**Algorithm 2** Sampling from DiffS2UT

**Input:** trained network parameters $\theta$ and K-means mapping function $g$, decoding steps $K$.

**Output:** generated sample $\mathbf{x}_0$.
Initialize $\mathbf{v}_T \sim \mathcal{N}(\mathbf{0}, \mathbf{I})$;
Convert continuous $\mathbf{v}_T$ to discrete space $\mathbf{x}_T = g^{-1}(\mathbf{v}_T)$;
Sample a subset $\{\tau_1, \ldots, \tau_K\}$ from the full diffusion trajectory $\{1, \ldots, T\}$;
**for** $t = \tau_1, \ldots, \tau_K$ **do**
    Draw $\hat{\mathbf{x}}_0 = \arg\max p_\theta(\mathbf{x}_t, t)$;
    Convert discrete $\hat{\mathbf{x}}_0$ to continuous space $\hat{\mathbf{v}}_0 = g(\hat{\mathbf{x}}_0)$;
    Draw $\mathbf{v}_{t-1} \sim q(\mathbf{v}_{t-1}|\mathbf{v}_t, \hat{\mathbf{v}}_0)$;
    Convert continuous $\mathbf{v}_t$ to discrete space $\mathbf{x}_t = g^{-1}(\mathbf{v}_t)$;
**end for**
**Return** $\mathbf{x}_0$.

---

units $\mathbf{x}_t = g^{-1}(\mathbf{v}_t) \in \{1, \ldots, K\}$ with the inversion function $g^{-1}$, which preserves the semantic structure of the speech representation. The perturbation $\mathbf{x}_t$ is semantically close to the target units $\mathbf{x}_0$ with the time step $t$ as control.

### 4.3 Sampling

The sampling details are described in Algorithm 2. Sampling in DiffSUT begins with a sequence comprising only continuous noisy vectors $\mathbf{v}_T \sim \mathcal{N}(\mathbf{0}, \mathbf{I})$. Then we convert $\mathbf{v}_T$ to discrete units $\mathbf{x}_T = g^{-1}(\mathbf{v}_T)$ with the inversion function $g^{-1}$. At each time step $t$, we first feed the noisy units $\mathbf{x}_t$ into the neural network and get the predicted target $\hat{\mathbf{x}}_0$ with $\arg\max$:

$$\hat{\mathbf{x}}_0 = \arg\max p_\theta(\mathbf{x}_t, t). \quad (7)$$

Since we apply the diffusion forward process in the continuous space, we need to convert the model prediction back to the K-means space $\hat{\mathbf{v}}_0 = g(\hat{\mathbf{x}}_0)$ with the K-means mapping function $g$. Then we sample $\mathbf{v}_{t-1}$ as in continuous diffusion models:

$$\mathbf{v}_{t-1} = \mathcal{N}(\mathbf{v}_{t-1}; \hat{\mathbf{v}}_0, \sigma_t^2 \mathbf{I}) \quad (8)$$

This strategy is more informative, as the noise is added to the semantically structured speech representation space, where the intensity is controlled by the time step $t$.

To accelerate the sampling speed, we pick a subset $\{\tau_1, \ldots, \tau_K\}$ from the full diffusion trajectory $\{1, \ldots, T\}$ for generation (Nichol and Dhariwal, 2021). Then a reverse step can be obtained by:

$$\mathbf{v}_{\tau_i} \sim q(\mathbf{x}_{\tau_{i-1}}|\mathbf{v}_{\tau_i}, \hat{\mathbf{v}}_0) \quad (9)$$

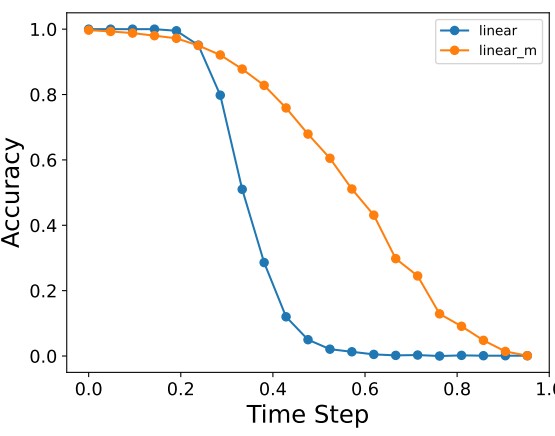

Figure 3: Accuracy of nearest neighbour search on the perturbation data in the K-means space.

Note that the conversion between the discrete and continuous spaces only requires $L_2$ distance computation, which can be accelerated with the FAISS library (Johnson et al., 2019) so that the time cost is negligible. Equipped with this method, we can execute the sampling algorithm more effectively.

### 4.4 Noise Schedule

Due to the fact that K-means is robust to small Gaussian noises, the unit sequence would not be corrupted when the added noise is small. As shown in Figure 3, the accuracy of the nearest neighbour search on the perturbation data remains 1.0 for the first 20% time steps with the linear noise schedule, which is useless since the model can directly

| Name | # Iter. | Es-En | Fr-En | En-Es | En-Fr | Param. | Speedup |
|---|---|---|---|---|---|---|---|
| **Cascaded systems:** | | | | | | | |
| S2T + tf TTS (Lee et al., 2022) | n.a. | 19.2 | 19.8 | 21.7 | 18.5 | n.a. | - |
| **S2UT systems with auto-regressive generation:** | | | | | | | |
| S2UT (Lee et al., 2022) | N | 13.1 | 15.4 | 16.4 | 15.8 | 71M | 1.0× |
| S2UT w/ ED (Duquenne et al., 2022) | N | 20.4 | 20.7 | 21.9 | 19.3 | 71M | 1.0× |
| S2UT w/ PT (Popuri et al., 2022) | N | 23.8 | - | 26.0 | - | 827M | - |
| **S2UT systems with diffusion generation:** | | | | | | | |
| S2UT-Absorbing (Austin et al., 2021) | 50 | 5.5 | 5.1 | 3.9 | 4.3 | 71M | 12.6× |
| S2UT-Multinomial (Hoogeboom et al., 2021) | 50 | 11.8 | 12.0 | 11.2 | 8.5 | 71M | 9.7× |
| DiffS2UT (Ours) | 5 | 13.5 | 14.1 | 13.8 | 12.9 | 71M | 14.4× |
| | 10 | 14.6 | 15.0 | 14.2 | 13.5 | 71M | 14.0× |
| | 20 | 14.8 | 15.1 | 14.4 | 13.6 | 71M | 12.4× |
| | 50 | **14.8** | **15.2** | **14.5** | **13.6** | 71M | 11.9× |

Table 1: BLEU scores achieved by training on VoxPopuli-S2S (Wang et al., 2021) training sets and evaluating on Europarl-ST (Iranzo-Sánchez et al., 2020) test sets. For the S2UT systems with diffusion generative models, the results are reproduced by ourselves.

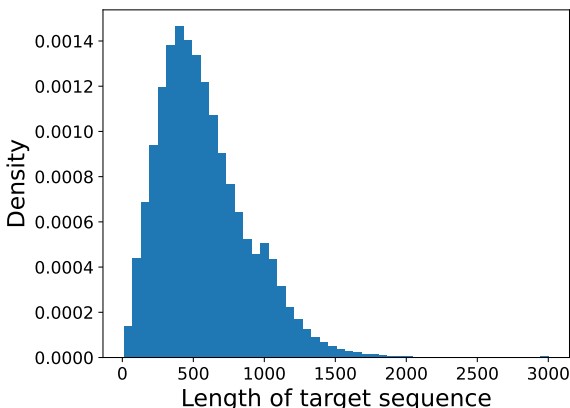

Figure 4: Statistics of the target sequence length in VoxPopuli-S2S Es-En training set.

copy the input as the target. To force the model to learn to recover the corrupted sequence, we propose a new noise schedule uniform for the DiffS2UT model. We initially set $\beta_0 = 0.3$ and decrease $1 - \bar{\beta}_t$ uniformly. The proposed uniform schedule improves the level of data utilization.

## 5 Experiments

### 5.1 Setup

**Datasets** We use the real-world dataset VoxPopuli-S2S (Wang et al., 2021) for model training. We examine four directions of language pairs: Spanish-English (Es-En), English-Spanish (En-Es), French-English (Fr-En), and English-French (En-Fr) on the test set from Europarl-ST (Iranzo-Sánchez et al., 2020). In order to avoid duplication with the corpus of the test set, we deleted the data before 2013 in

the training set. The statistics of the target unit sequence length in Es-En training set is shown in Figure 4. Different from the text translation data, the S2ST data is much longer with the average length larger than 500, inducing challenges to auto-regressive models with the long generation process.

**Evaluation** Following prior works (Lee et al., 2022), all speech outputs are decoded with the same open-sourced ASR models[1] to compute BLEU with respect to the reference transcribed translations using SACREBLEU (Post, 2018).

**Baselines** For comparison, we choose three S2UT baselines and reproduce two discrete diffusion models for S2ST. Among them, S2UT is the baseline model proposed in (Lee et al., 2022). S2UT w/ ED (Duquenne et al., 2022) trains the model on the mined S2ST datasets. S2UT w/ PT (Popuri et al., 2022) is initialized with the pre-trained wav2vec encoder and unit-based mBART decoder. We reproduce the absorbing (Austin et al., 2021) and multinomial (Hoogeboom et al., 2021) diffusion methods on the S2ST task, named S2UT-Absorbing and S2UT-Multinomial. The implementation details are described in Appendix A.1.

### 5.2 Results

Table 1 summarizes the results from different S2UT systems trained with VoxPopuli-S2S data (Wang et al., 2021). First, by comparing our proposed

---

[1] https://github.com/facebookresearch/fairseq/tree/main/examples/speech_to_speech/asr_bleu

| Models | BLEU |
|---|---|
| DiffS2UT | 14.8 |
| w/o K-means mapping | 12.1 |
| w/o beam size | 14.6 |

Table 2: The ablation study on the proposed methods. Results are conducted on the Es-En test set.

model with the models trained with conventional discrete diffusion processes (S2UT-Absorbing and S2UT-Multinomial), results show that our model achieves improvements of 10.0 and 3.0 in BLEU scores on the S2ST tasks. As shown in the results, existing discrete diffusion models struggle to perform well due to the complexity of the S2ST task over text translation. In comparison, our model can better utilize the semantic structures beneath the units through the K-means conversion. Furthermore, we also compare with the S2UT model based on auto-regressive generation, and our method is able to achieve comparable results with the baseline model S2UT even with only 10 decoding steps, which boosts the generation speed a lot.

### 5.3 Analysis

**On the Effect of Semantic Preserving Diffusion Process** We study the effects of the proposed components, which are listed in Table 2. It can be observed that without K-means mapping, DiffS2UT would degenerate to the multinomial diffusion and the BLEU score drops to 12.1, which is close to S2UT-Multinomial. We also study the effect of the predicted sequence length and ablate the beam search, we find that the model is slightly sensitive to it because the vocoder needs to predict the duration of each token which results in the time and speed of the final audio.

**On the Decoding Steps** We list the BLEU scores of different decoding step settings from 5 steps to 50 steps for all diffusion models in Table 1. We can see that the proposed DiffS2UT model achieves the BLEU score of 13.5 with 5 steps, which is even higher than the results of S2UT-Absorbing and S2UT-Multinomial with 50 steps. The results demonstrate the superiority of the proposed model DiffS2UT that it can surpass the conventional diffusion models with much fewer decoding steps.

**On the Intermediate Generation Results** To analyze the generation quality during the reverse process, we extract $\hat{x}_0$ at some of the reverse steps

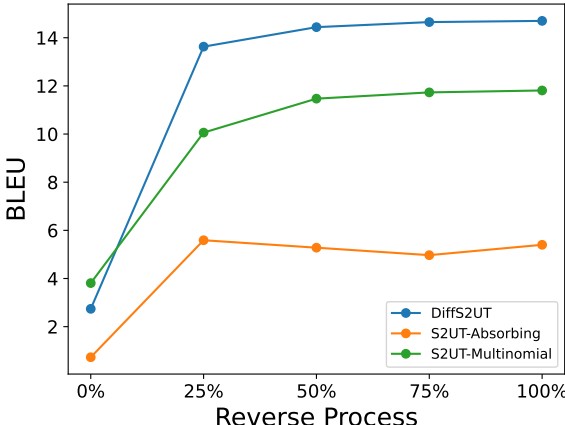

Figure 5: The BLEU score of intermediate $\hat{x}_0$ within a decoding process.

and evaluate the BLEU scores. The results are illustrated in Figure 5. All the diffusion models can achieve high BLEU scores in the first quarter of the reverse process and converge slowly in the rest steps. Among them, our DiffS2UT obtains the highest BLEU score in the first quarter, which is even higher than the final score of other diffusion models. According to the observation above, we can extract the intermediate $\hat{x}_0$ as the final decoding results to further accelerate the decoding speed with a negligible performance drop.

## 6 Conclusion

We present a diffusion generative model for the S2ST task named DiffS2UT that integrates the continuous and discrete diffusion models by decoupling the diffusion forward and backward processes. The model applies the diffusion forward process in the continuous speech representation space, while employing the diffusion backward process in the discrete speech unit space. In this way, we effectively preserve the semantic structure of the continuous speech representation space in the diffusion process. We evaluate the proposed DiffS2UT model on the real-world speech-to-speech datasets and demonstrate the significantly boosted generation quality compared to the conventional discrete diffusion models. Furthermore, the model can achieve comparable results to auto-regressive models but with much fewer decoding steps.

### Limitations

While the conversion between the continuous space and discrete space doses not add extra trainable model parameters, it does introduce some com-

putational overhead. The dimensionality of the K-means space is $D = 768$ and number of clusters is $K = 1000$. For each batch forward in training, the computation complexity $O(B \times L \times K \times D)$. Although we can accelerate the search process with the FAISS (Johnson et al., 2019) library without sacrificing the training speed, the GPU memory usage is still not negligible. Developing faster and more efficient nearest neighbor search tools remains an active area of research (Guo et al., 2020).

## Acknowledgements

This research was supported by the National Natural Science Foundation of China (Grant No. 62276245), and Anhui Provincial Natural Science Foundation (Grant No. 2008085J31).

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

## A Appendix

### A.1 Implementation Details

We follow the model architecture settings in (Lee et al., 2022), with a 12-layer speech encoder and 6-layer unit decoder with embedding size 512 and 8 attention heads. We train the model for 600k steps on VoxPopuli S2ST training datasets. We use Adam optimizer with learning rate 0.0003 for all translation directions, and inverse square root learning rate decay schedule with 10k warm up steps. To prevent overfitting, we use label smoothing of 0.2 for training. All S2UT systems except S2UT w/ PT (pre-trained models) are trained with an auxiliary task as (Lee et al., 2022). We find that sentence level knowledge distillation (Kim and Rush, 2016;

Gu et al., 2018) can boost the performance, so we adopt the typical method in the non-auto-regressive translation task with S2UT w/ ED as the teacher. We set diffusion steps $T = 1000$ for all diffusion based models with the linear noise schedule. In the sampling process, we use the length beam size 5 which means generating 5 candidates of different lengths at the same time and select the final prediction according to perplexity. All experiments are conducted using the FAIRSEQ toolkit (Ott et al., 2019). We use FAISS (Johnson et al., 2019) for fast nearest neighbor search in high-dimensional spaces.