# OpenReview forum: "DiffS2UT: A Semantic Preserving Diffusion Model for Textless Direct Speech-to-Speech Translation"
_EMNLP/2023/Conference — EMNLP 2023 Main_

### Official Review · Reviewer_2aZv · 2023-07-29

**Soundness:** 4

**Excitement:**

4: Strong: This paper deepens the understanding of some phenomenon or lowers the barriers to an existing research direction.

**Paper Topic And Main Contributions:**

The topic of this paper is textless direct speech-to-speech translation.
The authors propose an effective diffusion model for speech-to-speech translation.

**Reasons To Accept:**

The authors introduce diffusion generative models to the textless S2ST task.
The authors propose a novel diffusion framework that integrates the continuous and discrete diffusion models by decoupling the diffusion forward and backward processes.
Experiments demonstrate the proposed method.

**Reasons To Reject:**

The paper does not compare with other non-autoregressive methods, such as TranSpeech[1].
There still exists a certain gap between the method proposed in this paper and the auto-regressive methods.
[1] TranSpeech: Speech-to-Speech Translation With Bilateral Perturbation

**Reproducibility:**

4: Could mostly reproduce the results, but there may be some variation because of sample variance or minor variations in their interpretation of the protocol or method.

**Reviewer Confidence:**

3: Pretty sure, but there's a chance I missed something. Although I have a good feel for this area in general, I did not carefully check the paper's details, e.g., the math, experimental design, or novelty.

---

> ### Author Rebuttal · Authors · 2023-08-28
>
> We appreciate the valuable comments and constructive suggestions to help improve the paper. We give the response below.
>
> Q1: The comparison with TranSpeech.
>
> We did notice the TranSpeech model and have cited it in the related work. As introduced in Section 3.2, a direct textless S2ST system typically comprises three components: (1) The speech unit extraction model. (2) A translation model built upon units. (3) The unit to waveform model.
> Our proposed model, DiffS2UT, and the selected baselines all focus on the second component to build a stronger model for translation. In contrast, TranSpeech focuses on the first component to extract more robust speech units. Consequently, it would be not appropriate to take it as a baseline for comparison. It is noteworthy that our method and the selected baselines are fully compatible with the TranSpeech method. We can achieve further improvement with the TranSpeech method, which is beyond the scope of our focus though.
>
> Q2: The gap between DiffS2UT and auto-regressive models.
>
> Our approach achieves comparable results to the auto-regressive S2UT baseline, outperforming in some directions while lagging behind in others though. However, our primary objectives are to expedite speech generation and improve the generation quality over vanilla diffusion models. Therefore, it is not appropriate to solely assess our approach by comparing BLEU scores to auto-regressive models. Our experimental results clearly demonstrate superior efficiency compared to AR models (approximately 10 times faster) and improved generation quality when contrasted with vanilla diffusion models (achieving BLEU scores ranging from 3 to 5).

---

### Official Review · Reviewer_787n · 2023-08-07

**Soundness:** 2

**Excitement:**

2: Mediocre: This paper makes marginal contributions (vs non-contemporaneous work), so I would rather not see it in the conference.

**Paper Topic And Main Contributions:**

The paper introduces the diffusion model to direct speech-to-unit translation. Diffusion models, which have gained considerable success on other tasks in computer vision and natural language processing, construct forward and backward processing during training. The forward pass constructs a series of noisy sampling processes and backward recovers the original samples.

Specifically in the paper, the diffusion process is applied on the target discrete units. The author designs a scheme to add noise and reconstruct on the embedding level to solve issue of the discreteness of the units. The experiments show improvement over S2UT system with other diffusion-based model.

**Questions For The Authors:**

1. Line 515, how do you get the conclusion for semantic preserving? Table 2 only shows BLEU score, which is not a sufficient metrics for semantics.
2. Line 485, what is ED and PT from related work? I read the reference paper but still don't know the meaning of ED.

**Reasons To Accept:**

1. The idea in the paper is quite novel. The diffusion model has been introduced in many tasks and shows success. It is of great importance to show its potential to the speech-to-unit task.

2. The adaptation of the diffusion model on units is well designed. Instead of adding mask directly to discrete units, the author designed a scheme to move these operations to embedding space.

3. The paper gives a very comprehensive introduction to diffusion models and background.

**Reasons To Reject:**

1. The performance of the proposed methods is not good compared with the baseline. In the paper, the authors compare the proposed model with other S2UT model with diffusion model. However, the proposed model should be compared to a simpler yet much stronger baseline from Lee. et al 2021. Comparing with Lee. et al 2021, the proposed model only shows improvement on Es->En directions. There is no improvement or even downgrades for other directions.

2. The paper has too much introduction on the diffusion model and insufficient content of proposed model. The paper contains two pages of introduction section. The adaptation for diffusion to unit translation seems too abrupt and it would be good to have more analysis on the model behavior to show why actually the diffusion model helps the units generation.

**Reproducibility:**

3: Could reproduce the results with some difficulty. The settings of parameters are underspecified or subjectively determined; the training/evaluation data are not widely available.

**Reviewer Confidence:**

4: Quite sure. I tried to check the important points carefully. It's unlikely, though conceivable, that I missed something that should affect my ratings.

---

> ### Author Rebuttal · Authors · 2023-08-28
>
> We appreciate the valuable comments and constructive suggestions to help improve the paper. We give the response below.
>
> Q1: Lack of comparison with a baseline and the improvement of the proposed methods is minor.
>
> We do compare our approach with the auto-regressive baseline [Lee. et al] at the second line of Table 1. Our approach achieves comparable results to it, outperforming in some directions while lagging behind in others though. However, our primary objectives are to expedite speech generation and improve the generation quality over vanilla diffusion models. Therefore, it is not appropriate to solely assess our approach by comparing BLEU scores to auto-regressive models. Our experimental results clearly demonstrate superior efficiency compared to auto-regressive models (approximately 10 times faster) and improved generation quality when contrasted with vanilla diffusion models (achieving BLEU scores ranging from 3 to 5).
>
> Q2: Too much introduction to diffusion models and insufficient analysis regarding why the diffusion model helps the units generation.
>
> We will carefully revise the introduction section and try our best to reduce the length of the introduction while maintaining the article's clarity and coherence. We provide algorithms and implementation details of the proposed method in the paper. The diffusion model aims at helping accelerate the unit generation speed compared with auto-regressive models (about 10 times faster) and the incorporation of semantic structure improves the BLEU score compared to vanilla diffusion models.
>
> Q3: How do we get the conclusion for semantic preserving?
>
> It is hard to design a specific metric for semantic preserving. We give an intuitive explanation of semantic preserving in Figure 1(d), on which the motivation of DiffSUT is based. We also conduct the ablation study of semantic preserving, i.e. K-means mapping, in the analysis section. The main difference between DiffS2UT and vanilla diffusion models lies in semantic preserving, so we contend that the improvement of the BLEU score adequately substantiates its efficacy.
>
> Q4: The meanings of ED and PT.
>
> We explain the meanings of ED and PT in Section 5.1 Setup Baselines, which refer to Extra Data [1] and Pre-Training [2] respectively. We also introduce the related paper in the related work (line $220 \sim 226$). We will clarify it more clearly in the revised version.
>
> [1] Duquenne, Paul-Ambroise, et al. "SpeechMatrix: A Large-Scale Mined Corpus of Multilingual Speech-to-Speech Translations." arXiv preprint arXiv:2211.04508 (2022).
>
> [2] Popuri, Sravya, et al. "Enhanced direct speech-to-speech translation using self-supervised pre-training and data augmentation." arXiv preprint arXiv:2204.02967 (2022).

---

### Official Review · Reviewer_n2co · 2023-08-11

**Soundness:** 4

**Excitement:**

4: Strong: This paper deepens the understanding of some phenomenon or lowers the barriers to an existing research direction.

**Paper Topic And Main Contributions:**

While Diffusion Generative Models have achieved great success on image generation tasks, how to efficiently and effectively incorporate them into speech generation especially translation tasks remains a non-trivial problem. The authors propose a novel diffusion model by applying the diffusion forward process in the continuous speech representation space, while employing the diffusion backward process in the discrete speech unit space. The model achieves 14.8/15.2/14.5/13.6 BLEU score on the Eurapal-ST Es-En/Fr-En/En-Es/En-Fr test sets.

**Reasons To Accept:**

1. This is the first work that effectively introduces diffusion generative models to the textless S2ST task.
2. significantly surpassing the vanilla diffusion models, and achieving comparable performance to auto-regressive models while requiring much fewer generation steps.
3. The algorithm looks detailed.
4. The research is innovative, and is an exploratory study of diffusion generative models on text-free S2TS.

**Reasons To Reject:**

The introduction is too long and goes into too much detail.

**Reproducibility:**

4: Could mostly reproduce the results, but there may be some variation because of sample variance or minor variations in their interpretation of the protocol or method.

**Reviewer Confidence:**

2: Willing to defend my evaluation, but it is fairly likely that I missed some details, didn't understand some central points, or can't be sure about the novelty of the work.

---

> ### Author Rebuttal · Authors · 2023-08-28
>
> We appreciate the valuable comments and constructive suggestions to help improve the paper. We give the response below.
>
> Q1: The introduction is too long and goes into too much detail.
>
>  We will carefully revise the introduction section and try our best to reduce the length of the introduction while maintaining the article's clarity and coherence. We believe that it is crucial to introduce the direct S2ST task and continuous/discrete diffusion models meticulously and explain why it is important to integrate diffusion into S2ST, which includes substantial prior work that contributes to our concept development. Also, we believe it benefits both readers and the community to gain insight into the S2ST task and diffusion models.

---

### Official Review · Reviewer_1oEp · 2023-08-11

**Soundness:** 4

**Excitement:**

4: Strong: This paper deepens the understanding of some phenomenon or lowers the barriers to an existing research direction.

**Paper Topic And Main Contributions:**

This paper introduces diffusion models to the textless S2ST task, aiming to expedite the generation process. The authors propose integrating continuous and discrete diffusion models by disentangling the forward and backward diffusion processes. Experimental results demonstrate the effectiveness of the proposed method.

**Reasons To Accept:**

1. Drawing upon the one-to-one correspondence between discrete units and continuous K-means centroid vectors, this paper adeptly combines the continuous diffusion forward process with the discrete diffusion backward process. This idea is novel and reasonable to me.
2. The experimental results illustrate that the proposed method outperforms a direct application of vanilla discrete diffusion models.
3. The paper is well-written and easy to follow.

**Reasons To Reject:**

The authors assert that the diffusion-based architecture significantly expedites the generation process by comparing it to the auto-regressive generation step, which scales with audio length, in contrast to the fixed number of steps in diffusion generation. However, these two types of “generation step” have different meanings, which I think cannot be used to compare these two generation methods directly. A more straightforward comparison could involve fixing the target sequence length (e.g., 500, 1000, 1500…) and then evaluating the time and space consumption of different methods.

Considering the lower performance of the diffusion generation methods compared to the auto-regressive ones, I want to see the more intuitive comparison of the computational cost between different methods.

**Reproducibility:**

4: Could mostly reproduce the results, but there may be some variation because of sample variance or minor variations in their interpretation of the protocol or method.

**Reviewer Confidence:**

4: Quite sure. I tried to check the important points carefully. It's unlikely, though conceivable, that I missed something that should affect my ratings.

---

> ### Author Rebuttal · Authors · 2023-08-28
>
> We appreciate the valuable comments and constructive suggestions to help improve the paper. We give the response below.
>
> Q1: The comparison of generation steps.
>
> It is true that the generation steps for auto-regressive (AR) models and diffusion models have different meanings, where AR models generate at the token level and diffusion models generate at the sequence level. However, it is an advantage of diffusion models over AR models that the number of generation steps is fixed and does not increase linearly with the sequence length. In fact, the sequence length, which is equivalent to the generation steps in AR models, is close to the diffusion models. Since both models generate sentences with the same meaning, the difference in sequence length is negligible. Fixing the target sequence length for auto-regressive models is not recommended because the model implicitly generates the [EOS] token, which means it may stop generating before reaching the maximum sequence length or continue generating even after reaching it.
>
> Q2: The comparison of the computational cost.
>
> We conducted a benchmarking test following the official procedure in FAIRSEQ (examples/speech\_to\_speech/benchmarking) to compare the runtime costs of auto-regressive and DiffS2UT model (50 decoding steps) on the Es-En dataset, which is specifically designed for S2ST. As shown in the table below, DiffS2UT achieves a speedup of $11.9\times$, demonstrating its efficiency. We appreciate the suggestions and will include a computation cost comparison of all models to Table 1 in the revised paper.
>
> |Model|Mem/MiB|Runtime/sec|Speedup|
> |:------|:----------:|:-------------:|:---------:|
> |S2UT [Lee et al., 2022]   | $6293.73$  | $90.57$ | $1.0 \times$|
> |DiffS2UT (Ours)  | $8332.72$ | $7.61$  | $11.9 \times$|

---

### Meta-Review · Area_Chair_QX2s · 2023-09-16

**Recommendation:** 4

**Metareview:**

The paper proposed diffusion models for textless speech-to-speech translation. Proposes integrating continuous and discrete diffusion models.

Pros:
- Performance is surpassing vanilla diffusion models
- introduces diffusion generative models for textless s2st task

Cons:
- Comparison with other stronger baseline models is not done
- introduction is too long

---

### Decision · Program_Chairs · 2023-10-07

**Decision:**

Accept-Main

**Comment:**

The paper proposed diffusion models for textless speech-to-speech translation. Proposes integrating continuous and discrete diffusion models.

Pros:
- Performance is surpassing vanilla diffusion models
- introduces diffusion generative models for textless s2st task

Cons:
- Comparison with other stronger baseline models is not done
- introduction is too long